# Management of Inflammatory Bowel Disease in Patients with Current or Past Malignancy

**DOI:** 10.3390/cancers15041083

**Published:** 2023-02-08

**Authors:** Florian Poullenot, David Laharie

**Affiliations:** CHU de Bordeaux, Hôpital Haut-Lévêque, Service d’Hépato-Gastroentérologie et Oncologie Digestive, Université de Bordeaux, F-33000 Bordeaux, France

**Keywords:** IBD, cancer, previous cancer, incident cancer, immunosuppressant, anti-TNF

## Abstract

**Simple Summary:**

Physicians face therapeutic choices in an increasing number of IBD patients with current or past malignancy. Even if uncontrolled IBD may lead to severe complications, promoting cancer relapse through the use of immunomodulators is a concern. There is a lack of scientific evidence supporting the non-use of immunomodulators in IBD patients with previous cancer. Indeed, accumulative data suggest that the risk of recurrent and new cancer in patients with a history of cancer is not increased by thiopurines and anti-TNF agents. Most recently, cohort studies have found no differences in incident cancer rates in IBD patients with prior malignancy treated with vedolizumab or ustekinumab compared to those treated with anti-TNF agents. Therefore, decisions should be made on a case-by-case basis and shared by the oncologist and the patient, considering the natural history of cancer, the time elapsed since cancer diagnosis, and IBD prognosis.

**Abstract:**

Immunomodulators, conventional immunosuppressants, and/or biologics are used more often, earlier, and longer than before in patients with inflammatory bowel disease (IBD). Along with this, the lifetime risk for cancer is estimated to be 33% in the general population in Europe. Thus, physicians face therapeutic choices in an increasing number of IBD patients with current or past malignancy. Few data are available so far for managing this IBD subpopulation and this clinical concern still remains a critical situation for four reasons: (i) risk of reactivation of dormant micrometastasis with immunomodulators is of major concern, (ii) there is a knowledge gap about the safety of the most recent molecules, (iii) current guidelines do not recommend the use of immunomodulators within 2–5 years after a diagnosis of cancer, (iv) patients with previous cancers are excluded from clinical trials. There is a lack of scientific evidence supporting the non-use of immunomodulators in IBD patients with previous cancer. Indeed, accumulative data suggest that the risk for recurrent and new cancer in patients with a history of cancer is not increased by thiopurines and anti-TNF agents. Most recently, cohort studies have found no differences in incident cancer rates in IBD patients with prior malignancy treated with vedolizumab or ustekinumab compared to those treated with anti-TNF agents. Therefore, decisions should be shared by the oncologist and the patient, considering the natural history of cancer, the time elapsed since cancer diagnosis, and IBD prognosis.

## 1. Introduction

Inflammatory bowel diseases (IBDs) are lifelong disabling conditions affecting mainly young adults. Because treatment aims to control disease rather than merely symptoms [1], IBD therapy has moved to the earlier use of immunomodulators [2,3]—conventional immunosuppressants and/or biologics or small molecules. Greater cumulative exposure to these agents increases patients’ vulnerability to long-term adverse effects, including malignancies. Indeed, thiopurine use is associated with an increased risk for non-melanoma skin cancer (NMSC) and lymphoma [4,5,6,7], and anti-TNF use is associated with a higher risk for melanoma and lymphoma [8,9].

Because one-third to one quarter of the European population will develop malignancy [10], the frequency of patients with refractory IBD who may receive an immunomodulator while having cancer or having had it is increasing. Importantly, cancer survivors in the general population are at increased risk for new malignancy (overall relative risk 1.14), and the risk is markedly higher in patients who develop index cancer at a young age [11].

The management of IBD in a patient with prior cancer is critical. Even if uncontrolled IBD may lead to severe—sometimes life-threatening—complications, promoting cancer relapse through the use of immunomodulators is a concern [12,13]. There is a knowledge gap on the safety of IBD therapies in patients with prior malignancy, particularly with the most recently licensed drugs. To date, immunomodulators are not recommended within 2–5 years after a diagnosis of cancer [14], and patients with previous cancers are excluded from clinical trials [15]. Nonetheless, in patients with a history of cancer, data from large retrospective studies find no increased risk of relapse (or new cancer) with anti-TNFs and thiopurines [14,16,17,18,19], and data on vedolizumab or ustekinumab are reassuring [15,20,21]. Here, we review the literature on this topic to address practical messages for physicians in view of managing this critical situation.

## 2. Cancer and IBD: A Common Situation

IBD patients with cancer are frequently encountered in clinical practice. In a regional Danish IBD cohort diagnosed during 1978–2002 and followed until 2010, the overall cancer prevalence was 17% in 774 patients with Crohn’s disease (CD) and 14% in 1437 patients with ulcerative colitis (UC) [22]. In the Danish health care system, the standardised incidence ratios (SIRs)—which enable comparison of the cancer incidence in IBD patients with that in the general population—are 1.3 (95% confidence interval (95% CI) 1.2–1.4) in patients with CD and 1.1 (95% CI 1.0–1.1) in patients with UC. Absolute risks of cancers (according to tumour site) within 1–11 years following UC or CD diagnosis are shown in Figure 1 [23]. Therefore, the analysis of the relationship between IBD and cancer requires distinguishing cancers related to chronic intestinal inflammation, cancer related to IBD treatments, and unrelated malignancies [24].

### 2.1. Cancers Related to Chronic Inflammation

Chronic inflammation promotes carcinogenesis in IBD. This has mainly been shown with colorectal cancer (CRC) associated with UC and colonic CD. The cumulative incidence of colorectal cancer is around 5% at 30 years since IBD diagnosis [25]. In a recent large Scandinavian population-based cohort study that compared patients with or without UC, individuals with UC had an increased risk for CRC (HR 1.66; 95% CI 1.57–1.76) with an increased risk of death related to CRC (HR 1.59; 95% CI 1.46–1.72). These excess risks have declined substantially over time but remain significantly higher than in the general population [26]. Indeed, the IBD-related CRC risk has decreased, possibly related to improved control of digestive inflammation and/or more efficient colonoscopic screening. Nonetheless, colorectal carcinogenesis in IBD involves chronic inflammation [27]: the risk for CRC increases with disease duration, the anatomic extent of colitis, and the presence of other inflammatory disorders (such as primary sclerosing cholangitis), whereas it decreases after the use of anti-inflammatory drugs. Thiopurine and anti-TNF agents are associated with a decreased risk for CRC in patients with longstanding extensive colitis [28,29]. An unanswered question is whether increasing the number of available immunomodulators will prolong active inflammation (and therefore CRC risk) in patients previously eligible for coloproctectomy. Beyond CRC risk associated with IBD, other risks of digestive malignancy induced by chronic inflammation exist, particularly small bowel adenocarcinoma and ano-rectal cancer in patients with Crohn’s disease [30,31].

### 2.2. Cancers Related to IBD Treatments

Important safety concerns have been raised with regard to thiopurines and anti-TNF agents. IBD patients exposed to thiopurines are at higher risk for non-melanoma skin cancer, lymphoproliferative disorders, and urinary tract cancers [4,5,32]. IBD patients undergoing anti-TNF therapy are reportedly at higher risk for melanoma [6,8]. Anti-TNF monotherapy is associated with a small but statistically significant increased risk for lymphoma compared to exposure to neither medication; the risk is higher with combination therapy (adjusted HR 6.11; 95% CI 3.46–10.8) than with anti-TNF alone (adjusted HR 2.53; 95% CI 1.35–4.77) or thiopurines alone (adjusted HR 2.35; 95% CI 1.31–4.22) [9]. For patients with CD or UC, reassuring data on the risks of malignancy with ustekinumab, vedolizumab, and tofacitinib have been yielded by randomised trials that have excluded patients with cancer within the last 5 years [33,34,35]. However, real life data on these drugs in patients with IBD are sparse.

In other immune-mediated inflammatory diseases (IMIDs) such as psoriasis, ustekinumab exposure is not associated with an increased risk for malignancy compared to anti-TNF agents or nonbiologic therapies [36]. The data on risk for cancer in patients with rheumatoid arthritis under tofacitinib therapy are controversial. In a randomised, post-authorisation trial that included patients over 50 years old with rheumatoid arthritis and cardiovascular risk factors, the risk for cancer was higher with tofacitinib than with anti-TNFs [37]. This finding was not confirmed by a recent real-world series in rheumatoid arthritis [38]. A European prospective cohort study that assessed the safety and effectiveness of biologics in IBD provided insight on the safety of novel therapies such as new biologics and small molecules [39].

## 3. Cancer Risk in Patients with Previous Cancer

Patients with a history of cancer may develop local, regional, or metastatic recurrence, but also a new primary tumour in the same or another organ. It can be difficult to differentiate cancer recurrence from the development of a new tumour. To avoid confusion, in the medical literature, the term most used to define an oncological event following a history of cancer is incident cancer.

### 3.1. Cancer Risk in the General Population with Previous Cancer

Having a previous cancer increases the risk of having a new cancer by 14% (compared to the general population), and the risk is higher in young patients [11]. This increased risk is mainly explained by the fact that it mainly concerns patients with an oncological risk related to genetic susceptibility; moreover, exposure to radiotherapy or chemotherapy may promote pro-oncogenic effects [19].

### 3.2. Cancer Risk in IBD Patients with Previous Cancer

Consistent with what has been found in the general population, in the prospective French CESAME cohort (where 17,047 patients with IBD were enrolled from May 2004 to June 2005, and followed up until December 2007), the overall incidence of cancer was significantly higher in patients with than in those without previous cancer (multivariate-adjusted HR 1.9; 95% CI 1.2–3.0; *p* = 0.003); Figure 2 [19].

## 4. Risks of Immunomodulators in IBD Patients with Previous Cancer

As promotion of a cancer relapse by using immunomodulators is a concern, data from patients with organ transplants, who are severely immunosuppressed, could provide insight into immunomodulator safety in IBD patients with previous cancer. Penn et al. [40] proposed a registry of post-transplant cancers that included a classification of cancer recurrence risk according to tumoral site (Table 1). The majority of cancer recurrences are within 2 years of cancer diagnosis. The values are 33% from 2–5 years and 13% at >5 years after diagnosis [41]. This led to the recommendation of a cautious approach in patients with recent cancer. Indeed, tumour recurrence is more frequent immediately following malignancy, which coincides with the greatest consequences of lifting immunosurveillance.

The CESAME study generated data on the risk for cancer with conventional immunosuppressants, particularly thiopurine, in IBD patients with previous cancer. The study totalled 17,047 IBD patients, including 405 patients with a history of malignancy at inclusion. Among them, 23 patients developed incident cancer: 17 new cancers and 6 recurrences. The rates of new and recurrent cancers in the 312 patients who did not receive immunosuppressants at entry was 13.2/1000 patient-years (PY). In the 93 on immunosuppressants (77 on thiopurines and 10 on methotrexate), the rates of new and recurrent cancers were 6.0/1000 PY and 23.1/1000 PY and 3.9/1000 PY, respectively. There were no significant differences in the incidence of new or recurrent cancers between patients exposed or not exposed to immunosuppressants. These results were not substantially altered when non-melanoma skin cancers were excluded from the analysis [19]. Similarly, a meta-analysis of pooled data from all immune-mediated inflammatory diseases (IMIDs) [18] showed that conventional immunosuppressant use after an index cancer is not associated with an increased risk of recurrence or for new cancer. This was confirmed in a recent multicentre cohort study involving consecutive IBD patients with previous non-digestive malignancy [15]. However, physicians are reluctant to give conventional immunosuppressants to patients with previous malignancies. Indeed, in one study, among 80 patients with extra-intestinal cancer, those with cancer had comparable disease activity, but less use of immunomodulators (19% vs. 25%; *p* < 0.001) and an increased rate of surgery (4% vs. 2.5%; *p* < 0.05) compared to controls [42]. Very few data exist on methotrexate in patients with IBD and prior cancer. Nevertheless, this study also shows that it is often the therapeutic option chosen by experts when a medical option is retained.

## 5. Risks of Taking Anti-TNF Agents in IBD Patients with Previous Cancer

The proinflammatory cytokine TNF-alpha influences malignancy. On the one hand, TNF-alpha has an antitumour effect by initiating apoptosis. This has prompted a theoretical concern that inhibiting TNF-alpha may cause recurrence or rapid tumour progression in patients with cancer. On the other hand, TNF-alpha facilitates the survival and proliferation of neoplastic cells via the nuclear factor κB signalling pathway [43]. Thus, several phase 2 trials combining anti-TNF and chemotherapy have been conducted to see if blocking TNF-alpha could improve cancer-induced cachexia. No evidence of disease acceleration or increased mortality with infliximab has been identified in patients with refractory renal cell carcinoma, pancreatic adenocarcinoma, or non-small cell lung cancer [44,45,46]. Immune checkpoint inhibitors can improve the treatment of several cancers. These drugs increase T-cell activity and the antitumor immune response but also have immune-related adverse effects that affect the gastrointestinal tract in 7–30% of patients. The management of patients with intestinal adverse effects of immune checkpoint inhibitors should involve corticosteroids and the rapid introduction of infliximab for non-responders. To date, the use of infliximab in patients with active cancer has not been associated with a poorer oncological prognosis [47]. Expert opinions even go so far as to suggest to pursue anti-TNFs when starting immune checkpoint inhibitors in a patient with pre-existing IBD [48].

Because anti-TNFs are not recommended after a diagnosis of cancer, and because patients with previous cancers are excluded from clinical trials, the data on anti-TNF use in IBD patients with previous cancer are sparse. The supposed contraindication of anti-TNF therapy in IBD patients with a malignancy within the last 5 years is not supported by evidence. In one meta-analysis, anti-TNF use in IMID patients with previous cancer was not associated with an increased risk of recurrence or for new cancer [18]. The same result was reported in a Spanish registry of 520 IBD patients; the crude incident cancer rate in patients exposed to anti-TNFs after a prior cancer was 19.6 per 1000 PY [49]. This may be compared to 24.6 per 1000 PY in a multicentre study in the United States that analysed 333 IBD patients with prior cancer [50]. Two retrospective cohort studies [15,21] showed that anti-TNF therapy is not associated with an increased risk for new or recurrent cancer (crude incident cancer rates 42 and 34.5 per 1000 PY, respectively), in agreement with a study of anti-TNF agents in patients with rheumatoid arthritis [51,52]. Although limited by their retrospective design, these studies suggest that anti-TNF agents are safe in patients with previous cancer. As in the updated European Crohn’s and Colitis Organisation (ECCO) recommendations, anti-TNF agents may be used on the basis of multidisciplinary decisions involving oncologists, taking into consideration current and recent IBD activity and alternative treatment options [53].

## 6. Risk Related to Vedolizumab in IBD Patients with Previous Cancer

Due to its gut-selective mode of action, vedolizumab is of interest in patients with previous cancer—except for digestive cancers—because of the theoretically low risk for new cancer associated with its use. Being licensed more recently than anti-TNF agents, data on the risk for incident cancer induced by vedolizumab are sparse. Two retrospective studies of patients with IBD and a history of current or prior cancer took into consideration the cancer risk related to vedolizumab [15,21]. In one, patients with previous cancer were categorised into four cohorts (none, conventional immunosuppressant, anti-TNF, or vedolizumab) according to the first treatment to which they were exposed after the index cancer [15]. With a median follow-up duration of 55 months, 100 incident cancers were detected in the 538 patients. In this study, regarding the crude incidence rate calculation, follow-up time was calculated from the first administration of IBD therapy after index cancer to the last follow-up visit. For patients receiving no immunomodulator, during the follow-up period, follow-up started at the date of the index cancer. The crude cancer incidence rates per 1000 PY were 47.0 for patients receiving no immunomodulator, 36.6 for anti-TNFs, and 33.6 for vedolizumab (*p* = 0.23). Incident-cancer-free survival rates were not different between patients on anti-TNFs and those on vedolizumab (*p* = 0.56) (Figure 3). The other compared 96 patients exposed to vedolizumab after a prior diagnosis of cancer to 184 and 183 patients exposed to anti-TNF or no immunosuppressive therapy, respectively [21]. In a multivariable Cox model, after adjusting for confounders, there was no increase in the risk for new or recurrent cancer after vedolizumab (HR 1.38; 95% CI 0.38–1.36) or anti-TNF (HR 1.03; 95% CI 0.65–1.64) therapy compared to foregoing the use of an immunomodulator.

The most recent ECCO guidelines state that IBD patients with a history of malignancy do not appear to have an increased risk for cancer recurrence when treated with vedolizumab [53]. Therefore, this could be a therapeutic option in selected situations when treatment of IBD is required despite a recent history of non-digestive cancer.

## 7. Risk Related to Ustekinumab in IBD Patients with Previous Cancer

Ustekinumab, an anti-interleukin 12/23 agent, is used to treat IBD and selectively targets immune pathways. There are reassuring data on the risk for subsequent cancer that suggest that ustekinumab can be used in patients with previous cancer. In a retrospective study of 390 patients with IBD and a history of cancer, there was no increase in new or recurrent cancer rates with vedolizumab (adjusted HR 1.36; 95% CI 0.27–7.01) or ustekinumab (adjusted HR 0.96; 95% CI 0.17–5.41) [20] use (Figure 4). Of note, the number of patients exposed to ustekinumab was small (14 patients including 2 who developed incident cancer). A retrospective study of IBD patients reported similar results [54]. However, the small sample size may have hampered the detection of small differences in subsequent cancers. Such a low level of evidence precludes recommendations for use of ustekinumab in patients with IBD and malignancy [53].

## 8. Risk Related to Tofacitinib in IBD Patients with Previous Cancer

To the best of our knowledge, in IBD and in other IMIDs, there are no data on the risks related to tofacitinib and other Janus-kinase inhibitors in patients with previous cancer. Based on this knowledge gap and the increased risk for cancer with tofacitinib in elderly rheumatoid arthritis patients, these agents should be avoided in patients with previous cancer requiring treatment for IBD [37,38].

## 9. Recommendations in Clinical Practice

Treatment of IBD patients after a diagnosis of cancer requires a case-by-case approach. Decisions must be made by both the oncologist and the patient, taking into consideration the natural history of cancer, the time elapsed since cancer diagnosis, and the IBD prognosis.

In the ECCO guidelines presented at the 17th Congress of the ECCO from 16–19 February 2022, five statements were made on the management of IBD in patients with a history of recent or active cancer [53]:ECCO statement 20: Current evidence suggests that there is no additional risk of incident cancer with thiopurine use in patients with IBD and a past history of malignancy, above that known to be associated with this class. However, most observational data are from patients starting treatment with thiopurines more than 5 years after cancer resolution, and in patients with a low risk of cancer recurrence: (evidence level (EL) 4) consensus 100%.ECCO statement 21: Thiopurines should preferably be withdrawn in patients with an active cancer diagnosis (EL4). Patients with non-aggressive basal cell carcinoma or preneoplastic lesions of the cervix may continue thiopurine therapy if they are closely monitored: (EL5) consensus 100%.ECCO statement 22: There are insufficient data to make recommendations regarding the safety of the use of methotrexate in patients with prior malignancies: (EL5) consensus 100%.ECCO statement 23: In patients with IBD and current or previous cancer, anti-TNF agents may be used. However, data on individual cancer types and timing of anti-TNF treatment are lacking (EL4). Multidisciplinary decisions should be made on a case-by-case basis involving oncologists, considering factors including current and recent IBD activity and alternative treatment options: (EL5) consensus 100%.ECCO statement 24: IBD patients with a history of malignancy do not appear to have an increased risk of cancer recurrence when treated with vedolizumab (EL4). There are insufficient data regarding the safety of vedolizumab in patients with active malignancy. There is insufficient evidence to make recommendations on the use of ustekinumab or JAK inhibitors for patients with current or prior malignancy (EL5). Multidisciplinary decisions should be made on a case-by-case basis involving oncologists, considering factors including current and recent IBD activity and alternative treatment options: (EL5) consensus 100%.

These statements have limited evidence and will be updated in the future. However, they can assist gastroenterologists and oncologists caring for patients with IBD and previous or current cancer. An algorithm summarising management guidelines was proposed in a review published almost 10 years ago [43]. We propose an updated version of that algorithm to account for recent data (Figure 5). Within 2 years of completing cancer treatment, 5-aminosalicylates, corticosteroids, antibiotics, nutritional therapy, surgery, and possibly methotrexate should form the basis of IBD treatment [24]. In patients with refractory IBD, administration of anti-TNF as monotherapy, vedolizumab, and possibly ustekinumab should be decided by the gastroenterologist, the oncologist, and the patient. In patients with a cancer diagnosis 2–5 years prior and refractory IBD, anti-TNF or vedolizumab monotherapy can be considered when therapy is resumed.

## 10. Conclusions

As IBD treatment moves to prolonged courses of immunomodulators, and as the IBD population ages, an increasing number of patients with prior malignancy may receive conventional IS and/or biologics and/or small molecules. In all cases, the decision to stop or to resume IBD therapy in a patient with previous or current cancer should be carefully evaluated. A multidisciplinary case-by-case approach is key, and emphasis should be placed on the individual risk of cancer recurrence, the potential risk posed by each immunosuppressant drug in the setting of that cancer, and the time elapsed since completion of cancer therapy.

## Figures and Tables

**Figure 1 cancers-15-01083-f001:**
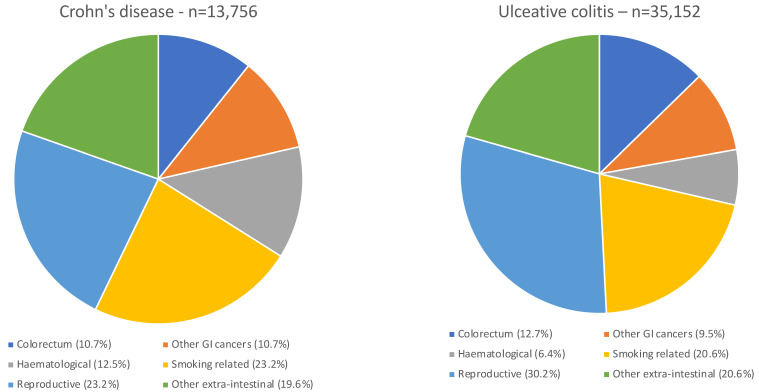
Cancers in a Danish nationwide population-based cohort study of patients with Crohn’s disease and ulcerative colitis with 30 years of follow-up [23].

**Figure 2 cancers-15-01083-f002:**
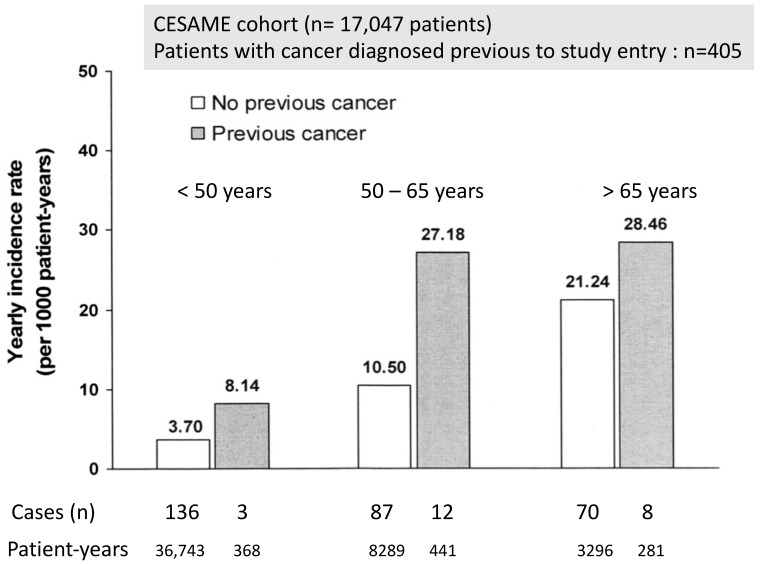
Annual incidence rate of cancer in IBD patients with and without previous cancer [19].

**Figure 3 cancers-15-01083-f003:**
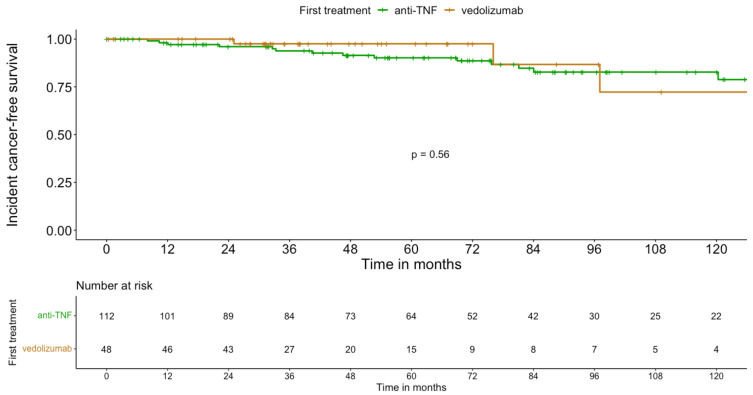
Incident-cancer-free survival with anti-TNF and vedolizumab in IBD patients with previous malignancy [15].

**Figure 4 cancers-15-01083-f004:**
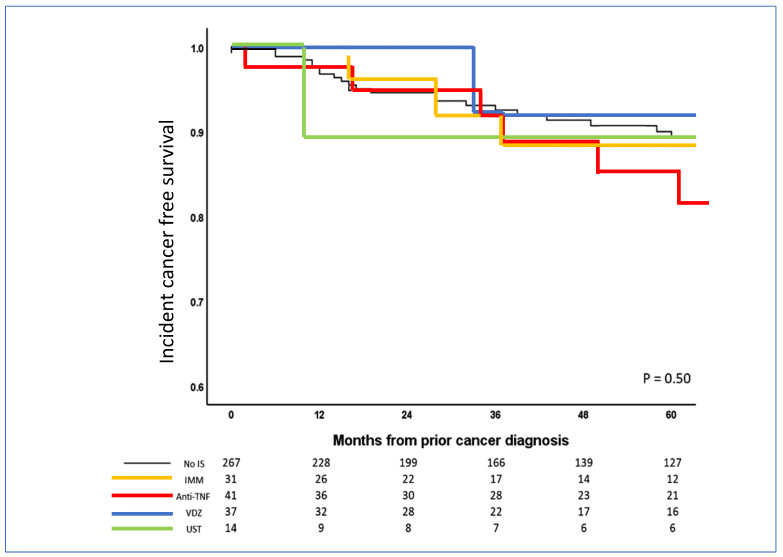
Incident-cancer-free survival with immunosuppressants, anti-TNFs, vedolizumab, and ustekinumab in IBD patients with previous malignancy [20].

**Figure 5 cancers-15-01083-f005:**
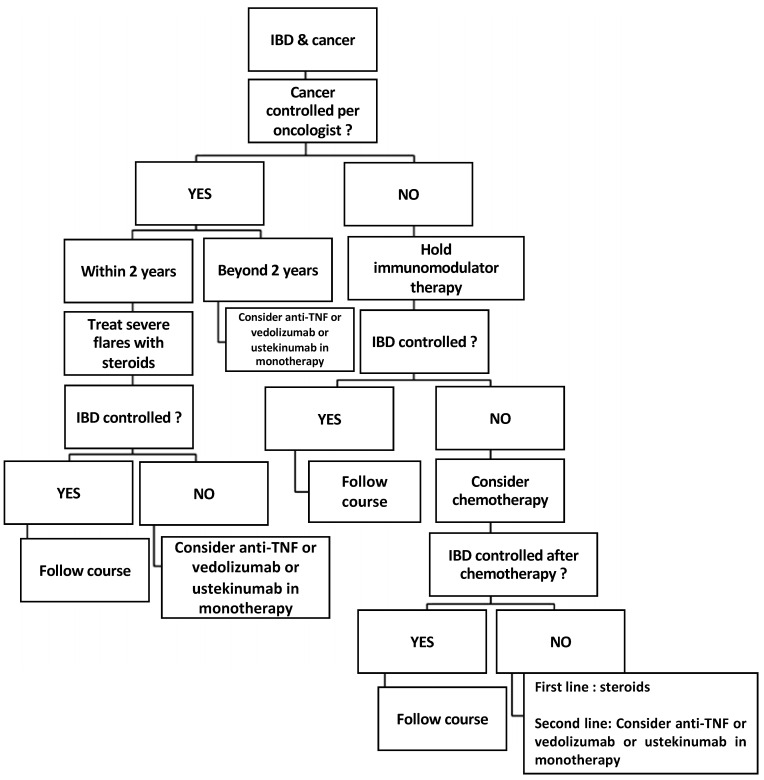
Updated algorithm for managing IBD patients with previous or current malignancy (adapted from reference [43]).

**Table 1 cancers-15-01083-t001:** Risk of recurrence of pre-existing cancer under post-transplant immunosuppressive therapy (adapted from Penn et al. [40]).

Risk Level	Organ
Low (<10%)	Kidney (asymptomatic) Lymphoma Testicle Uterine cervix Thyroid
Intermediate (11–25%)	Breast Uterine body Colon Prostate
High (>25%)	Bladder Kidney (symptomatic) Sarcoma Melanoma Non-melanoma skin cancer Myeloma

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
