# Peer review of "Management of Inflammatory Bowel Disease in Patients with Current or Past Malignancy"

_cancers, 2023, doi:10.3390/cancers15041083_

Round 1
Reviewer 1 Report
In this review article, the authors F. Poullenot and D. Laharie aimed to share scientific evidence supporting that the risk for recurrent and new cancer in IBD patients with a history of cancer is not increased by treatments of anti-TNF agents or thiopurines; Treatments of vedolizumab or ustekinumab have no impact oil incident cancer rates in IBD patients with prior malignancy compared to those treated with anti-TNF agents.
Overall, I found the review article very interesting and holds potential. However, I have several serious concerns/comments, as listed below:
Minor:
- I suggest adding line numbers in the manuscript draft. It was very difficult to keep track of my notes while reading.
- In the Abstract section - line 6, “the knowledge gap … from clinical trials”, could you please rephrase the sentence? The whole sentence does not make sense as the subject shifted from the “knowledge gap” to “the fact” and then to “because patients.” Similar errors happen several times throughout the article, which is very common for non-native speakers. I suggest the authors have a professional editor to assist them while revising the article.
- Figure 1 - I strongly recommend that the authors change the color scheme. Right now, it is extremely difficult to distinguish different groups.
- Figure 4 - Same comment as above. I strongly recommend that the authors change the color scheme. Right now, it is extremely difficult to distinguish different groups.
- Page 8 - Recommendations in clinical practice - when “ECCO” first appears in the article, it is recommended to include the full name. This abbreviation could mean something else for readers who do not live in Europe.
Major:
- Figure 1 - Two groups are missing/not indicated in the figure legend; Sample sizes need to be added to the figure legend.
- Figure 2 - why is there no in-text reference to Figure 2? The authors need to add an in-text reference and provide an interpretation of Figure 2. Is there a significant difference between the “No previous cancer” and the “Previous cancer” group? Does age contribute to the difference? What are the sample size and the source of data?
- Figure 3 - could the authors please explain how the “incident cancer-free survival” is calculated? The table with values provided below Figure 3 does not reflect what is presented in the figure. And what statistical method was used to get the p-value (0.56)? This should be included in the discussion.
- Figure 4 - same comment as above. Could the authors please explain how the “cancer-free survival” is calculated? The table with values provided below Figure 4 does not reflect what is presented in the figure.
- Inappropriate citation - references 15, 16, 18, 24, 43, and 48 are written by the same authors of this article - F. Poullenot and D. Laharie:
- Reference 15 appears murilple times throughout the article (pages 2,3,6,and 7) as the single evidence/literature support. I strongly recommend that the authors find alternative references to support the statements/conclusions made based on reference 24; otherwise, the credibility of this article would be significantly lowered. Specifically, on page 7, line 2, the authors first introduced two studies that they would like to discuss in the following content - references 19 and 21. However, in the following sentences, the authors cite reference 15, then reference 21. Is this mistake? Please double-check that you are citing the correct article.
- Reference 18 (first appears on page 6, line 3) is a meta-analysis published in 2016. Generally speaking, narrative reviews, systematic reviews, or meta-analyses are based on original research articles and are considered secondary sources. Therefore, they are not recommended to be used in a systematic review. I suggest that the authors directly cite the original research articles to support their discussion instead of citing the meta-analysis published by themselves.
- References 16, 43, and 48 are OK to keep.
- Reference 24 appears multiple times throughout the article (pages 2,4,5,6, and 9) as the single evidence/literature support. I strongly recommend that the authors find alternative references to support the statements/conclusions made based on reference 24; otherwise, the credibility of this article could be significantly lowered.
Author Response
Reviewer: 1
Minor:
- I suggest adding line numbers in the manuscript draft. It was very difficult to keep track of my notes while reading.
Answer: It has been done in the revised manuscript
- In the Abstract section - line 6, “the knowledge gap … from clinical trials”, could you please rephrase the sentence? The whole sentence does not make sense as the subject shifted from the “knowledge gap” to “the fact” and then to “because patients.” Similar errors happen several times throughout the article, which is very common for non-native speakers. I suggest the authors have a professional editor to assist them while revising the article.
Answer: In the Abstract section – line 6, the sentence has been rephrased. Nonetheless, The English in this document has been ever checked by at least two professional editors, both native speakers of English. For a certificate, please see:
http://www.textcheck.com/certificate/gDy4Hd
- Figure 1 - I strongly recommend that the authors change the color scheme. Right now, it is extremely difficult to distinguish different groups.
Answer : Thank you for this suggestion. Color scheme has been changed.
- Figure 4 - Same comment as above. I strongly recommend that the authors change the color scheme. Right now, it is extremely difficult to distinguish different groups.
Answer: Thank you for this suggestion. Color scheme has also been changed.
- Page 8 - Recommendations in clinical practice - when “ECCO” first appears in the article, it is recommended to include the full name. This abbreviation could mean something else for readers who do not live in Europe.
Answer: The full name was included when first appears.
Major:
- Figure 1 - Two groups are missing/not indicated in the figure legend; Sample sizes need to be added to the figure legend.
Answer: Thank you for this comment. The two missing groups have been added. Sample sizes have been added.
- Figure 2 - why is there no in-text reference to Figure 2? The authors need to add an in-text reference and provide an interpretation of Figure 2. Is there a significant difference between the “No previous cancer” and the “Previous cancer” group? Does age contribute to the difference? What are the sample size and the source of data?
Answer: Thank you for this comment and sorry and sorry for forgetting to reference this figure in the text. We add an in-text reference. Figure 2 shows incidence rates of incident cancer according to age at entry into the cohort and previous history of cancer in the CESAME cohort (from Beaugerie L, Carrat F, Colombel JF, Bouvier AM, Sokol H, Babouri A, et al. Risk of new or recurrent cancer under immunosuppressive therapy in patients with IBD and previous cancer. Gut. sept 2014;63(9):1416‑23). As mentioned in the text, there is a significant difference between the “previous cancer” and the “No previous cancer” group (multivariate-adjusted HR 1.9; 95% CI 1.2–3.0; p = 0.003). In the CESAME cohort, 17 047 patients were prospectively enrolled from May 2004 to June 2005, and followed-up until December 2007. We add this information in the manuscript.
- Figure 3 - could the authors please explain how the “incident cancer-free survival” is calculated? The table with values provided below Figure 3 does not reflect what is presented in the figure. And what statistical method was used to get the p-value (0.56)? This should be included in the discussion.
Answer: During the follow-up period, incident cancer was defined as recurrence of the known cancer or occurrence of a new cancer in the reference 15. This way of defining incident cancers is the same in all the studies cited; we have clarified this specific point in our manuscript. In the reference 15, crude cancer incidence rates were compared between the cohorts and, next, patients from the anti-TNF and vedolizumab cohorts were matched on age, lymph node and metastasis extension and malignancy recurrence risk using a propensity-score analysis with a 1:1 ratio. Adjusted cancer incidence rates were compared using the chi-square test. For survival analyses, time-dependent Kaplan–Meier analysis for survival curves and the log-rank test for univariate comparisons were used. The table with values below Figure 3 shows the number of patients concerned in each group according to time.
- Figure 4 - same comment as above. Could the authors please explain how the “cancer-free survival” is calculated? The table with values provided below Figure 4 does not reflect what is presented in the figure.
Answer: As above, in the reference 20, the time to incident cancer was calculated from the date of initial cancer diagnosis to the date of the first new or recurrent malignancy. The cancer free survival is “the survival without incident cancer”. We changed the figure by replacing “cancer-free survival” by “Incident cancer free survival”. The table with values below Figure 4 shows the number of patients concerned in each group according to time.
- Inappropriate citation - references 15, 16, 18, 24, 43, and 48 are written by the same authors of this article - F. Poullenot and D. Laharie:
Reference 15 appears murilple times throughout the article (pages 2,3,6,and 7) as the single evidence/literature support. I strongly recommend that the authors find alternative references to support the statements/conclusions made based on reference 24; otherwise, the credibility of this article would be significantly lowered. Specifically, on page 7, line 2, the authors first introduced two studies that they would like to discuss in the following content - references 19 and 21. However, in the following sentences, the authors cite reference 15, then reference 21. Is this mistake? Please double-check that you are citing the correct article.
Reference 18 (first appears on page 6, line 3) is a meta-analysis published in 2016. Generally speaking, narrative reviews, systematic reviews, or meta-analyses are based on original research articles and are considered secondary sources. Therefore, they are not recommended to be used in a systematic review. I suggest that the authors directly cite the original research articles to support their discussion instead of citing the meta-analysis published by themselves.
References 16, 43, and 48 are OK to keep.
Reference 24 appears multiple times throughout the article (pages 2,4,5,6, and 9) as the single evidence/literature support. I strongly recommend that the authors find alternative references to support the statements/conclusions made based on reference 24; otherwise, the credibility of this article could be significantly lowered.
Answer : Thank you very much for this comment but the assistant editor of Cancers asked us to carry out a review of the literature based on what has already been published and therefore on some of the studies we have already published on this complex subject beyond the field of gastroenterology. Meta-analysis on this topic are sparse and that’s why we decide to cite them (and because they give an overview including rheumatology and dermatology). We confirm a mistake on page 7 (corrected now) and we delete some citations coming from reference 15 and 24.
Reviewer 2 Report
THis is a useful contribution for clinician guidance on a topic which is often contentious.
Minor issues to correct:
1. Page 5: "As promote cancer relapse.." sentence construction needs attention eg "As promotion of cancer relapse..."
2. "casuistic" is not the correct word. THis has theological implications.
3. The complex sentence at the end of page 5 "he rates of new and recrrent cancers in 313 patients..." should be simplified by splitting into two sentences, especially as it is the only reference to methotrexate.
4. The position of methotrexate in the Guidance needs clarification. The ECCO statement is guarded about its use, and it does not appear in Figure 5. However, there methotrexate is included amongst the "safe" recommendations in the Guidance provided by the authors. There needs to be more consideration of this, especially given the ECCO hesitation
5. Page 6. What is IMID?
6. Page 5. Table 1 - the Format of Table 1 needs attention to differentiate the intermediate and high risk sections
Author Response
Reviewer 2:
Minor issues to correct:
- Page 5: "As promote cancer relapse.." sentence construction needs attention eg "As promotion of cancer relapse..."
Answer: Thank you for this comment. We corrected the sentence.
- "casuistic" is not the correct word. THis has theological implications.
Answer: Thank you for this comment. We replaced by “case-by-case approach”.
- The complex sentence at the end of page 5 "he rates of new and recrrent cancers in 313 patients..." should be simplified by splitting into two sentences, especially as it is the only reference to methotrexate.
Answer: We changed the sentence. Thank you for this comment.
- The position of methotrexate in the Guidance needs clarification. The ECCO statement is guarded about its use, and it does not appear in Figure 5. However, there methotrexate is included amongst the "safe" recommendations in the Guidance provided by the authors. There needs to be more consideration of this, especially given the ECCO hesitation.
Answer: Thank you for this comment. We agree with the reviewer. We tuned-down the Guidance provided at the end of manuscript, and we added a comment in the section about immunosuppressants. Indeed, the study of Rajca et al [42] shows that methotrexate is often the therapeutic option chosen by experts when medical option is retained but it’s a “real” expert opinion and most of the data are with thiopurines.
- Page 6. What is IMID?
Answer : IMID is Immune Mediated Inflammatory Diseases. We added it in the manuscript.
- Page 5. Table 1 - the Format of Table 1 needs attention to differentiate the intermediate and high risk sections
Answer: Thank you for this comment. We changed it.
Round 2
Reviewer 1 Report
Thank you for sending me the revised manuscript and responding to all my comments. I appreciate the authors' corrections and revisions and am glad to suggest that this manuscript may be accepted in its present form.
Author Response
Thank you for your answer